# Long-Term Fate and Efficacy of a Biomimetic (Sr)-Apatite-Coated Carbon Patch Used for Bone Reconstruction

**DOI:** 10.3390/jfb14050246

**Published:** 2023-04-26

**Authors:** Florian Olivier, Christophe Drouet, Olivier Marsan, Vincent Sarou-Kanian, Samah Rekima, Nadine Gautier, Franck Fayon, Sylvie Bonnamy, Nathalie Rochet

**Affiliations:** 1CNRS, Université d’Orléans, ICMN UMR 7374, 45071 Orléans, France; sylvie.bonnamy@cnrs-orleans.fr; 2CIRIMAT, Université de Toulouse, CNRS/UT3/INP, 31062 Toulouse, France; christophe.drouet@cirimat.fr (C.D.); olivier.marsan@toulouse-inp.fr (O.M.); 3CNRS, Université d’Orléans, CEMHTI UPR 3079, 45071 Orléans, France; vincent.sarou-kanian@cnrs-orleans.fr (V.S.-K.); franck.fayon@cnrs-orleans.fr (F.F.); 4Université Côte d’Azur, INSERM, CNRS, iBV, 06107 Nice, France; samah.rekima@univ-cotedazur.fr (S.R.); nadine.gautier@univ-cotedazur.fr (N.G.); nathalie.rochet@univ-cotedazur.fr (N.R.)

**Keywords:** bone reconstruction, activated carbon fiber cloth, Sr^2+^-doped apatite, osteointegration, long-term fate, Raman microspectroscopy

## Abstract

Critical bone defect repair remains a major medical challenge. Developing biocompatible materials with bone-healing ability is a key field of research, and calcium-deficient apatites (CDA) are appealing bioactive candidates. We previously described a method to cover activated carbon cloths (ACC) with CDA or strontium-doped CDA coatings to generate bone patches. Our previous study in rats revealed that apposition of ACC or ACC/CDA patches on cortical bone defects accelerated bone repair in the short term. This study aimed to analyze in the medium term the reconstruction of cortical bone in the presence of ACC/CDA or ACC/10Sr-CDA patches corresponding to 6 at.% of strontium substitution. It also aimed to examine the behavior of these cloths in the medium and long term, in situ and at distance. Our results at day 26 confirm the particular efficacy of strontium-doped patches on bone reconstruction, leading to new thick bone with high bone quality as quantified by Raman microspectroscopy. At 6 months the biocompatibility and complete osteointegration of these carbon cloths and the absence of micrometric carbon debris, either out of the implantation site or within peripheral organs, was confirmed. These results demonstrate that these composite carbon patches are promising biomaterials to accelerate bone reconstruction.

## 1. Introduction

Reconstruction of large bone defects after trauma or pathological conditions remains a challenge in orthopedics and maxillo-facial surgeries [1]. Moreover, the aging of the world’s population is contributing to a significant increase in the number of patients requiring bone repair [2,3]. To address this challenge, many synthetic bone substitutes have been developed and used, including in particular calcium phosphates (CaPs) such as stoichiometric hydroxyapatite (HA), biphasic calcium phosphate (BCP), or other calcium phosphate phases [4,5,6,7,8,9]. CaP compounds such as apatites exhibit a chemical composition close to natural bone mineral, which involves ca. 70 wt.% of apatitic phase [10], thus highlighting their intrinsic biocompatibility. Another advantage of using CaPs as bone substitutes, especially with bone-like apatites, is the ease with which their structure can be modified to confer new biological properties [11,12]. Calcium ions, in particular, may be substituted at least in part by a wealth of other cations, including bioactive ones such as strontium ions [13,14,15,16,17], despite a somewhat larger ionic radius (Ca^2+^ = 1.00 Å and Sr^2+^ = 1.18 Å). Partial substitution of Ca^2+^ by Sr^2+^ in CaP biomaterials, e.g., between 3 and 7 at.%, was shown to stimulate pre-osteoblast activity and differentiation, while at the same time inhibiting osteoclast activity and differentiation [18,19,20], thus promoting bone tissue growth over bone resorption while altering neither the mineralization process nor the chemical composition of newly formed bone [21,22,23]. In animal models, the effectiveness of Sr-enriched biomaterials in stimulating bone formation/remodeling and in improving the quality of the newly formed bone have been clearly evidenced [24,25].

In addition to their use as bone fillers, biomimetic CaP compounds can be used as coatings to improve the osteointegration of various implant types [26]. Several methods have been proposed to deposit such bioactive coatings such as atmospheric plasma spraying (APS), physical and chemical vapor deposition (PVD and CVD) [27], sol-gel [28], electrodeposition [29], and by immersion in a biomimetic solution [30]. For porous materials (3D), electrodeposition has been described as among the most promising methods since it can deposit homogeneous bioactive coatings such as CaP both on the interior and exterior surfaces of conductive scaffolds, regardless of sample size and pore geometry [31].

In line with this we have previously demonstrated that the sono-electrodeposition process facilitates the attainment of a biomimetic calcium-deficient apatite (CDA) coating on activated carbon fiber cloth (ACC) to be used as bone patches [32]. Moreover, we established that this CDA coating could be easily doped with Sr^2+^ ions (Sr-CDA) [15,17]. Further experiments in vitro revealed that strontium exhibited a positive and dose-dependent effect on osteoblast activity and proliferation, with optimal results obtained with the ACC/10Sr-CDA patch corresponding to 6 at.% of strontium substitution [17,33]. More recently, we implanted three different patches (ACC, ACC/CDA and ACC/CDA-drug) in a rat model of a large cortical bone defect with the view to follow short-term in vivo outcomes. We showed that ACC/CDA patches were efficient biomaterials for cortical bone healing at 7 and 14 days post-surgery. Overall, these studies demonstrated a double role of this composite biomaterial: (1) a guiding effect of ACC on bone formation, and (2) an accelerating effect of CDA on bone reconstruction [34]. We also demonstrated the possibility of using this system to deliver, in situ, a drug in the first days after surgery [35].

The goal of the present study was threefold. Our first objective was to investigate the efficacy of the composite carbon fiber patches in vivo in the medium term (26 days post-surgery). Using a model of critical bone defect in rat femurs, we analyzed bone reconstruction in the presence or absence of patches of ACC/CDA or the strontium-doped equivalent ACC/10Sr-CDA. This allowed us to test the significance of strontium doping. The second objective of this work was to analyze the medium- and long-term fate of these patches (26 days and 6 months after surgery), i.e., their biocompatibility and osteointegration. The third objective was to answer the question of whether carbon debris could be found out of the implantation site and within peripheral organs. Analyses of ACC and CDA coatings (undoped or Sr^2+^-doped) were carried out by microscopy techniques (optical and scanning electron microscopy, SEM). The quality of the newly formed bone was checked using Raman microspectroscopy through the analysis of different quantitative parameters (mineral-to-organic ratio, apatite maturity state, carbonation level, hydroxyproline-to-proline ratio) and compared to those obtained with native cortical bone.

## 2. Materials and Methods

### 2.1. Materials

The ACC/CDA and ACC/10Sr-CDA patches were prepared by sono-electrodeposition following the protocol described previously [17]. The flexible ACC used was referred to as FM50K (Zorflex^®^ ACC, Calgon Carbon Corporation, Pittsburgh, PA, USA). Patches of 10 × 4 mm were cut and rinsed cautiously in deionized water for 15 s to remove any undesirable carbon residue. They were then dried at 70 °C under vacuum for 12 h. Before surgery, patches were sterilized by dry heat at 180 °C for 2 h.

### 2.2. Study Design

All animal procedures obtained the approval of the local animal health care committee (Authorization reference numbers APAFIS#11442-2017091413241724 and APAFIS#19215-2019021512224201). The experimental design of this work is described in Figure 1. This model of bilateral femoral defect in rats was firstly used to analyze the effect of the ACC/CDA and ACC/10Sr-CDA patches on bone defect reconstruction, secondly to study the medium- and long-term fate of ACC/CDA patches, and thirdly to look for the putative presence of micrometric carbon debris outside of the implantation site and within peripheral organs. In the first experiment, 18 male Lewis rats weighing 358 ± 14 g (Janvier Labs, Le Genest-Saint-Isle, France) were operated upon and the 36 femurs were randomized into three groups (Table 1) corresponding to ACC/CDA, ACC/10Sr-CDA and control (no patch). The effect of the patches was analyzed after 26 days. In the second experiment, 3 male Lewis rats weighing 313 ± 9 g (Janvier Labs, Le Genest-Saint-Isle, France) were operated upon and the 6 femurs were treated with ACC/CDA patches. Histological analyses of the patches and of various tissues/organs from these animals were performed after 6 months.

### 2.3. Surgical Procedure

Anesthesia was performed by inhalation of isoflurane (5%) followed by IM injection of ketamine (30 mg/kg, Virbac, Carros, France) and xylasine (7.6 mg/kg, Dechra, Montigny-le-Bretonneux, France), SC injection of an antimuscarinic (0.1 mg/kg robinul-V, Magny-vernois, France) and IP injection of an opiate (Buprecare^®^, Axience, Pantin, France). During surgery the animals were kept under a mixture of air (5 L/min) and oxygen (2 L/min).

The surgical protocol was conducted as described in our previous study [34]. A bilateral cylindrical defect of 2.7 × 4–5 mm (diameter × depth) including the periosteum was created at the femoral metaphysis. Twelve defects were covered with a ACC/CDA patch, twelve with a ACC/10Sr-CDA patch and twelve were not covered. A post-operation X-ray (Faxitron, Edimex, Le Plessis Grammoire, France) check was performed—at a voltage of 35 kV and an exposure time of 9.5 s—to verify the absence of a fracture.

### 2.4. Post-Surgery Care

During the first week post-surgery the rats were housed in individual cages and monitored daily for complications. They were then grouped in threes and monitored thrice weekly. Analgesics and an antibiotic were administered in the drinking water for the first 6 days after surgery as detailed previously [34].

### 2.5. Micro-Computed Tomography—µCT

In the first experiment, all the animals were been killed after 26 days with CO_2_. After dissection and fixation in formalin (10% buffered), the whole femurs were scanned using a high-resolution µCT scanner (Skyscan 1173, Synergie4, Evry, France) as described previously [34]. The DataViewer software (1.5.6.2 version, Bruker microCT, Billerica, MA, USA) and the CT Analyser software (1.17.7.2 + version, Bruker microCT, Billerica, MA, USA) were used to define the region of interest, to observe the shape and to measure the thickness of newly formed bone. Furthermore, a numerical scoring was used to analyze bone shape following Table 2 indications. The Matlab software (R2017b version, The MathWorks Inc., Natick, MA, USA) was used to quantify the newly formed mineral. The region containing the bone defect was defined inside a constant volume cylinder (3D). Each 2D slice of this volume was then binarized individually to select the newly formed mineral. The resulting binary images, acting as masks, were multiplied by their own slices to finally calculate the volume of the newly formed mineral. The ratio between the new mineral volume and the initial volume of the cortical defect was calculated and designated as “bone mineral filling %”.

### 2.6. Statistical Analysis

Statistical evaluation of data was performed using Student’s test. The experiment was repeated twelve times and the results presented are the mean of the values. Data are reported as ± standard deviation at a significance level of at * *p* < 0.05 and ** *p* < 0.01.

### 2.7. Scanning Electron Microscopy—SEM

After dissection of the femur, the patches were gently separated from the bone surface with razor blade and stored in formalin. They were then mounted on aluminum stubs and sputter coated with 2 to 3 nm gold layer. Examination was performed using a field emission scanning electron microscope (SEM-Hitachi S4500, Hitachi High-Tech Corporation, Tokyo, Japan) operating at 5 kV.

### 2.8. Raman Microspectroscopy

The quality of the newly formed bone was analyzed by Raman microspectroscopy on a representative explant at day 26 post-surgery (non-histologically stained sample) and corresponding to the implantation of the Sr-doped CDA-coated ACC patch (ACC/10Sr-CDA) with complete closure of the bone defect. With this aim, thick sections were cut in the transverse plane in the middle of initial bone defect with a circular diamond blade.

The analyses were carried out using a confocal Raman Labram HR 800 Horiba Jobin Yvon microscope (Horiba FRANCE SAS, Palaiseau, France). Spectra were collected in the range of 400—1800 cm^−1^ giving rise to characteristic signatures from both bone mineral and organic components [36]. The samples were exposed to continuous laser radiation supplied by a continue laser at λ = 633 nm with a power of 8 mW. The specimen under analysis was placed under an Olympus BX 41 microscope (Olympus Europa SE & CO, Hamburg, Germany) and the focused at a long distance ×50 objective with a numerical aperture (NA) of 0.750, providing a lateral resolution of 1.03 µm (=1.22 × λ/NA) and an axial resolution of 4.5 µm (=4 × λ/NA^2^). Punctual analyses were made along a line starting at the opening of the bone defect, at the place of apposition of the ACC patch, denoted by a depth level Z = 0, up to a depth of 800 µm with data points recorded every 200 µm. For the sake of comparison, the spectral signature of the native cortical bone opposite to the defect was also analyzed. To control the positioning of the analyzed domain, an XYZ motorized table (Märzhäuser GMBH & Co., Wetzlar, Germany) with an accuracy of 0.1 µm was employed. The use of an autofocus system whose amplitude range was optimized to the roughness of the area studied allowed us to adjust the focus of the laser.

A silicon standard enabled the equipment to be frequency-calibrated using the 1st order line of silicon at 520.7 cm^−1^ with an accuracy of ±1 cm^−1^.

The spectrum of each point was acquired using a 600 gr/mm grating with a spectral resolution of 1 cm^−1^, and collected with a quantum well detector cooled to −60 °C by double Pelletier effect (CCD Synapse). Each spectrum was acquired with a 30 s time and 3 accumulations. Data processing was performed using the Labspec 6 software.

### 2.9. Histology

Histological analyses were performed after 26 days in the first experiment and after 6 months in the second. After fixation, the femurs were partially decalcified in 10% (*w*/*v*) ethylene diamine tetra-acetic acid (EDTA, Alfa Aesar, Thermo Fisher Scientific, Illkirch, France) solution for 8 weeks. Bone blocks of 5 mm including holes and patches were then cut and embedded in paraffin. For each block 2 sections of 10 µm were made every 100 µm from the beginning to the end of the blocks. All sections were stained with hematoxylin and eosin (HE, Hematoxylin (Harris Hematoxylin solution, Sigma-Aldrich, Saint Quentin Fallavier, France)—Eosin (Eosin 0.5% alcoholic, Diapath, Voisins-le-Bretonneux, France) and slides were scanned using a whole slide scanner, Vectra Polaris (Akoya, Villebon-sur-Yvette, France).

Additionally, at 6 months post-surgery, histological analyses were performed on several key organs, namely the brain, heart, lungs, kidney, liver and spleen, after fixation in 10% buffered formalin. These organs were cut into small cubes of 5 mm and embedded in paraffin. For each cube, between 8 and 12 sections of 7 µm were made, stained with HE and scanned. Using the HALO Image Analysis Platform (Indica Labs, Albuquerque, NM, USA), we trained the classifier module using artificial intelligence to segment carbon features versus tissue regions on control slides. The trained algorithm was then run on all the slide sections to detect the presence of carbon debris. Resolution was estimated to be below 1 µm (Appendix A).

## 3. Results

### 3.1. Bone Reconstruction at Day 26 Post-Surgery

All the animals were sacrificed at day 26. µCT analyses showed that the amount of bone mineral newly formed in the defects was similar in the three groups (ACC/CDA, ACC/10Sr-CDA and control, Figure 2A). However, the segmental quantification of total bone volume at the left limit, in the center and at the right limit of the defect revealed a statistically significant increase in thickness in the center of the reconstructed bone in the presence of ACC/10Sr-CDA compared to ACC/CDA and control groups (Figure 2B). The average increase in new bone volume in the presence of ACC/10Sr-CDA patches was estimated to be +30%. Taken together, these results indicate that the amount of new mineral matrix, although quantitatively similar in the three groups, occupied a larger volume after the application of strontium-doped patches. This strongly suggests that the bone formed in contact with the strontium-doped patches has a larger porous structure than that of the bone formed in the other two groups.

From the µCT images, the shape of all the femurs was scored according to the scheme described in Table 2. The average scores for bone reconstruction shape were, respectively, 1.9, 2.6 and 3 in the control, ACC/CDA and ACC/10Sr-CDA groups (Figure 2C). More precisely, in the presence of ACC/10Sr-CDA, 5 out of 12 femurs had reached a native bone shape compared to 3/12 in the ACC/CDA group and 2/12 in the control group (Figure 2D).

The physicochemical quality of the newly formed bone was studied in detail by implementing Raman microspectroscopy analyses on a characteristic explant (ACC/10Sr-CDA group) for which the reconstruction was completed. Figure 3A shows the typical spectrum of the native cortical bone opposite the bone defect, along with the assignment of the main spectral contributions either relative to the mineral or to the organic components. Figure 3B shows the positioning of the data points analyzed, which gave rise to the corresponding spectra, normalized with reference to the apatite peak denoted “ν_1_(PO_4_)” at 960.2 ± 0.2 cm^−1^ and reported in Figure 3C. For completeness, the reference spectrum of the starting (non-coated) ACC patch is also shown in this figure to unveil the potential positioning of initial organic contributions.

Remarkable spectral contributions of the mineral may be found typically at positions centered, in this study, at 962.2 ± 0.2 cm^−1^ for ν_1_(PO_4_), 439.4 ± 1.9 cm^−1^ for ν_2_(PO_4_), 581.3 ± 2.0 cm^−1^ for ν_4_(PO_4_) and 1071.9 ± 0.5 cm^−1^ for ν_1_(CO_3)_. However, the asymmetric shape of the intense ν_1_(PO_4_) band cannot be reproduced with a single Gaussian/Lorentzian peak; it can be more accurately simulated with two components: one positioned at ~950.0 ± 4.0 cm^−1^ (ν_1_(PO_4_)_am_ attributable to an amorphous part and a second contribution at 961.3 ± 1.3 cm^−1^ attributable to the vibration mode ν_1_(PO_4_)_apatite_ from the crystalline domains of apatite [37]). The data point corresponding to Z = 400 µm is given in Figure 3E as illustrative example.

Beside the spectral signature of bone mineral, the organic component (essentially collagen) also gave raise to specific signatures, such as the amide III bands in the region 1220–1320 cm^−1^, but also CH_2_ bands (at 1449.6 ± 4.5 cm^−1^ and 1462.9 ± 6.3 cm^−1^) [38,39] and peaks assignable to amino acid residues such as hydroxyproline (877.4 ± 1.9 cm^−1^), proline (853.5 ± 1.9 cm^−1^) or phenylalanine (1003.7 ± 0.7 cm^−1^). As several spectral contributions were often superimposed, a curve-fitting strategy was used (as exemplified on Figure 3E) to determine each contributing peak area as much as possible (note that amide II bands may somewhat superimpose with ν_3_(CO_3_), and that the ν_2_(CO_3_) carbonate vibration may also to some extent alter the band at 877.4 ± 1.9 cm^−1^ essentially assigned to hydroxyproline, for example). Appendix A (as Appendix A) reports the characteristics of each peak for each compound of interest, along with the assignments of the most remarkable bands that will be discussed in the following.

Among the key parameters to follow is the mineral-to-organic ratio [40]. Indeed, in the regular scenario of bone formation in vivo, the first stage of formation of a collagen matrix is undergone through activity of osteoblasts, which get mineralized in a subsequent step. Thus, the quantification of the mineral-to-organic ratio may be seen as a way to evaluate the evolution of the bone maturation state [41]. In addition, the mechanical properties of the bony matrix were found to be directly linked to this mineral-to-organic ratio among other features [42,43,44,45]. In this work, we evaluated this ratio from the analysis of the ν_1_(PO_4_) band of the mineral centered at 960.2 ± 0.2 cm^−1^, and the amide III bands (~1220–1320 cm^−1^) from the organic component, mainly collagen. Our results are based on the analysis of the ν_1_(PO_4_)/amide III ratio point as the mineral-to-organic proportion close to the native cortical bone, except for the point localized at Z = 0 where the ratio found was decreased by 60%. Considering all the internal data points up to Z = 800 µm, despite some variability, the mean value of 6.2 ± 0.9 compares well with that of the native cortical bone evaluated at 6.1. Alternatively, since the Raman bands located at wavenumbers greater than 1150 cm^−1^ can essentially be attributed to the organic component (amide and CH_2_ vibrations), we also calculated the intensity ratio between ν _1_(PO_4_) and the sum of all contributions in the range of 1150–1550 cm^−1^. Again, the point corresponding to Z = 0 fell to about 50% beneath the “reference” value of the native bone (2.3), while all other data points led to a mean of 1.7 ± 0.2. Globally, these mineral-to-organic quantifications show the low mineralization state of the outermost point at the initial level of patch apposition (Z = 0), while the other more internal analyses indicated a degree of mineralization rather close (seemingly slightly lower) to the native cortical bone value.

A second parameter related to the apatite maturation state was also investigated. Again, it was expected that a more mature bone apatite phase would be associated with a higher degree of crystallinity [46] and lower amount of amorphous components (either on the surface hydrated layer on the nanocrystals or by the presence of amorphous calcium phosphate precursor phase often observed in vivo in early mineralization stages). The degree of apatite maturity can thus be evaluated at least qualitatively by Raman spectroscopy through the position and Full Width at Half Maximum (FWHM) of the ν_1_(PO_4_) band. Indeed, the position of the ν_1_(PO_4_) maximum tends to shift toward larger wavenumbers as crystallinity increases, and in parallel the FWHM tends to decrease due to a greater structural order. In the present study, interestingly we observed a very constant value for the position of the ν_1_(PO_4_) band at 960.2 ± 0.2 cm^−1^ for all data points from Z = 0 to 800 µm, very close to the value 960.1 cm^−1^ found for the native cortical bone. Similarly, the FWHM of this band reached a mean value of 19.9 ± 2.7 cm^−1^, very close to the 19.6 cm^−1^ found for the cortical bone reference. In addition, in order to assess the degree of maturity of the bone mineral phase, we also determined another parameter (not particularly addressed in the literature around bone maturation after implanting a bone substitute, to the best of our knowledge), namely the ratio between the contribution of the amorphous domains in bone mineral with respect to the overall ν_1_(PO_4_) band intensity. As mentioned above, we noticed that the ν_1_(PO_4_) band shows a rather asymmetrical and wide shape (as usually observed in bone) which can only be accounted for by two contributions. While the low wavenumber contribution (centered at 950.0 ± 4.0 cm^−1^) was attributed to the amorphous part, the second contribution at 961.3 ± 1.3 cm^−1^ was assigned to the crystalline domains. Despite some of the variability among the data points, no specific trend could be evidenced and the mean value of the amorphous-to-total phosphate ratio (0.33 ± 0.19) remained rather close to the value of 0.37 of the cortical bone.

A third parameter regarding bone apatite aging in vivo is an increase in carbonation [47]. The CO_3_ level may however also depend on other factors such as whether the type of bone is cortical or trabecular [47]. Here, evaluation of the carbonation level was assessed from the area ratio between the ν_1_CO_3_ band at 1071.9 ± 0.5 cm^−1^ and the ν_1_(PO_4_) band, leading to a mean of 0.17 ± 0.11 compared to 0.28 for the cortical bone. Although no clear trend could be seen along the depth of the analyzed domain, these results suggest a lower degree of carbonation for the newly- formed bone tissue. With regard to the above results, it therefore seems that the overall degree of mineralization was highly comparable to that of the native cortical bone but with a lower carbonation level of the apatitic mineral phase. It may be presupposed that this degree of carbonation would continue to progressively increase up to the mean value of cortical bone during the course of bone aging.

Beside changes observed in the mineral phase, the organic component of the newly formed bone tissue should also be examined as collagen can also undergo modifications upon aging/neoformation, which may lead to relevant changes in Raman spectra. In particular, the modalities of collagen crosslinking may progress over time [48]. It was shown in the literature that an increase in hydroxylation of some amino acid residues occurred in collagen upon aging. In particular, the hydroxylation of proline into hydroxyproline has been reported upon mineralization progression [36,49,50], indicative of collagen post-translational modifications. In turn, the increased exposure of –OH hanging groups likely modifies the H-bonding network between collagen fibers and thus potentially their 3D organization and potential interaction with the mineral. We have here evaluated the hydroxyproline-to-proline ratio by calculating the intensity ratio between the hydroxyproline band at 877.4 ± 1.9 cm^−1^ and the proline band at 853.5 ± 1.9 cm^−1^. It may however be remarked that there is high uncertainty on the curve-fitting strategy as the two contributions are rather close to each other and largely overlapping. Furthermore, additional peak contributions may also interfere in this spectral region. No specific trend could be unveiled, and the mean value of the hydroxyproline-to-proline ratio approached 0.55 ± 0.36 compared to 0.26 for cortical bone. Taking into account the high standard deviation and absence of monotonous trend of this parameter, it is difficult to draw conclusions on any significant difference in collagen hydroxylation rate between the newly formed bone and reference cortical bone. We expect local variation in the mechanical properties of the newly formed bone tissue, due in part to the porosity of the new bony matrix in formation, and also local variations in the mineral-to-organic ratio among other parameters. Evaluation at the micro-scale of mechanical properties by approaches such as Brillouin–Raman microspectroscopy could prove helpful in the future to explore this type of new bone tissue further [43,44,45].

As a conclusion to this Raman microspectroscopy survey, we may thus state that the bone tissue formed at day 26 exhibits physicochemical characteristics that are very close to regular cortical bone, with no significant difference, either in the organic component or in the mineral component (except a somewhat lower degree of carbonation of the apatite phase). It may also be remarked that the spectral features of the three groups of biomaterials tested in vivo—namely control, ACC/CDA and ACC/10Sr-CDA—remained very similar, as may be seen in Appendix A.

### 3.2. Patch Fate at 26 Days and 6 Months Post-Surgery

#### 3.2.1. Macroscopic and Microscopical Observation of the Patches after 26 Days

On day 26 after surgery, at the time of dissection of the femurs, no macroscopic signs of inflammation were detected in the surrounding tissue of the limb. The patches were in place, strongly cohesive to the bone surface and covered by a connective tissue (Figure 4, red arrow). After gentle removing, we found that the colonized patches had lost their initial flexibility and kept the shape of the underlying bone.

In order to check whether the structure of the carbon fibers had been modified after 26 days of implantation, SEM observations were carried out to compare uncoated ACC fibers with CDA and 10Sr-CDA-coated ACC fibers after 26 days. The original knitting of uncoated carbon fibers is depicted in Figure 5A and their original lobed shape is shown in Figure 5B. Figure 5C,D show the structure of CDA-coated (5C) and 10Sr-CDA-coated (5D) carbon fibers before implantation. Figure 5E corresponds to the cross section of a fiber, showing the surface of the carbon fiber (red dotted line) and the surface of the CDA coating (white dotted line). These pictures demonstrate that the presence of the CDA coating hides the natural lobed shape of the fibers (red dots), leading to a final circular-like, non-lobed surface (white dots). At day 26 post-surgery, SEM analyses revealed that the carbon fibers had recovered their original lobed shape and were no longer coated with CDA or 10Sr-CDA (Figure 5H,I, red outline). Taken together, these results suggest that the deposit was completely resorbed by this timepoint. These pictures also show that both ACC/CDA and ACC/10Sr-CDA patches were colonized by biological tissue (Figure 5F,G).

#### 3.2.2. Histological Analysis of ACC/CDA and ACC/10Sr-CDA Patches after 26 Days of Implantation

Optical microscopy observations revealed similar results for ACC/CDA and ACC/10Sr-CDA patches at day 26 (Figure 6A,B). A thorough examination showed that the ACC fibers were fully colonized by an active fibroblastic tissue, including fibrillar collagen fibers, many fibroblastic cells, blood vessels and giant multinucleated cells.

#### 3.2.3. Histological Analysis of ACC Patches after 6 Months of Implantation

The long-term fate of the patches was analyzed 6 months after surgery (Figure 7). The same histological protocol as before was applied for the preparation of the samples. Bone blocks including the defect and the patch were entirely cut and the patches were examined along their entire length. This revealed that the long-term fate of the carbon cloth depended on its location, either facing the defect (Figure 7(B1)) or at distance from it (Figure 7(A1)). Optical microscopy examination showed that the results were markedly different depending on these two locations. As shown in Figure 7A, at distance from the defect the carbon fibers were colonized by a fibroblastic tissue similar to that observed at day 26 but with more fibrillar collagenous trabeculae and a lower number of multinuclear giant cells (Figure 7(A4)). Conversely, examination of the patches in the region facing the bone defect revealed that the carbon fibers were completely colonized by a mature lamellar bone including osteocytes, numerous blood vessels and bone marrow islets. This demonstrated that the carbon fibers located over the defect were fully osteointegrated within the surrounding native cortical bone (Figure 7(B3–B5)).

#### 3.2.4. Histological Analysis of Organs

The possible migration of carbon fiber debris outside the site of implantation at 6 months post-surgery was also studied by histology. The potential presence of carbon microparticles was examined in different major organs, namely the heart, liver, lungs, spleen, kidney and brain.

Automatic analyses of histologic slides were performed in order to quantify the three main regions, namely the blank, tissue and carbon areas (Figure 8). ACC patch and ACC patch/bone were used as positive controls (purposefully containing carbon fibers) to train the classifier module using artificial intelligence to segment carbon features versus tissue regions (Figure 8A, ACC, ACC/Bone and Appendix A). As expected and shown in Figure 8B, ACC and ACC/Bone positive controls were composed of tissue areas (Figure 8B, pink), blank areas (40 and 50% of the surface for each control, respectively) (Figure 8B, white), and carbon fibers (18.4% and 7.1%, respectively), (Figure 8B, black). Our results show that all the organs examined using this algorithm were composed exclusively of tissue and blank areas (Figure 8A,B). No carbon microparticles were detected in any of the organs after 6 months post-surgery.

## 4. Discussion

This work was conducted to investigate the efficacy and the fate of activated carbon fiber cloths (ACC) coated with calcium-deficient apatite (CDA), either doped or not doped with Sr^2+^ cations.

Our in vivo data in rats at day 26 post-surgery clearly indicate that the apposition of these composite patches improves the bone healing process more effectively in the presence of a strontium-doped CDA coating. These data confirm the guiding effect of the ACC fiber patches that we described previously [34] as well as the osteogenic effect of strontium cations described in many studies [18,19,20]. Moreover we also showed that this increase in new bone volume in the presence of strontium occurred, although the amount of new formed mineral matrix was similar in the three groups, namely control, ACC/CDA and ACC/10Sr-CDA. Despite the fact that we could not make an accurate measurement of porosity, we can make the hypothesis that new bone formed in the presence of an Sr^2+^-doped patch has a higher porous structure than new bone formed in the presence of undoped patches or in the absence of a patch. This higher porous structure should likely provide a better interconnection supporting cell migration, neo-vascularization, new tissue ingrowth and bone healing. Our results further showed that this significant augmentation of bone volume in the presence of Sr^2+^ was related to an increase in thickness at the center of the reconstructed bone. This result is consistent with our bone shape scoring data showing that ACC/10Sr-CDA patch apposition led to the highest number of femurs recovering their native convex shape. This is also consistent with a recent work of Quade et al. regarding the boosting role of Sr^2+^ ions in BMP-2-mediated bone regeneration of mice femurs [25]. In this study, the authors concluded, by means of morphological scores, that the quality of the newly formed bone was significantly enhanced in Sr-modified scaffolds in a murine segmental bone defect model.

The quality of the newly formed bone was further assessed via an in-depth Raman microspectroscopy analysis. Indeed, vibrational spectroscopies (FTIR, Raman) have been shown to be especially well-suited to the analysis of several key physicochemical parameters related to the mineral and organic components of the bony matrix [40]. In particular, Raman spectra can provide both qualitative and quantitative information on the collagen and the apatite moieties, without being significantly overwhelmed by the contribution of water molecules which are, in contrast, highly active in infrared spectroscopy. Although several vibrational signatures of different chemical groups may partially overlap in the Raman spectra, a detailed analysis of the spectra using curve fitting routines provides band positions, widths, and relative intensities of specific vibrational modes that can be used to follow the bone reconstruction process. Upon regular aging, the bone tissue undergoes several types of changes that in turn generate spectral modifications. It is reasonable to suppose that similar trends may also be expected after implantation of a bone substitute material or any device favoring bone ingrowth/regrowth from a bone defect, as is the case in the present study.

A significant part of this study was also dedicated to examine the fate of the implanted biomaterial over time. At day 26 post-surgery as well as after 6 months, macroscopic examination revealed no inflammatory reaction and no sign of patch rejection. Neither migration nor degradation of ACC was observed and the patches were found to be strongly adhesive to the bone. After dissection, we noted that they were rigid and had kept the shape of the host bone; the patches perfectly matched the bone surface morphology, allowing for a strong contact with the femoral diaphysis. SEM observation at 26 days unveiled the native lobed form of uncoated carbon fibers, thus suggesting the dissolution of the CDA coating. We can hypothesize that the CDA and Sr^2+^ CDA coatings were progressively resorbed over time, and that its dissolution may have participated in the bone regeneration process through precursor phosphate, calcium and strontium ions. This resorption process is probably a consequence of the presence of the numerous osteoclast-like/giant cells, as confirmed by the histological analyses.

Histological analyses at both time points, i.e., 26 days and 6 months, confirmed the high biocompatibility of the ACC patches which were entirely colonized by collagen tissue. At both timepoints, neo-vascularization was a major indicator of good integration as well as the presence of giant multinucleated cells. These cells are most likely osteoclast-like cells whose presence is consistent with the first phase of bone formation by osteoblasts, as observed in the presence of all bone biomaterials implanted in vivo. Furthermore, analysis at 6 months revealed that the fibers facing the defect were fully osteointegrated in the newly formed cortical bone. The presence of bone marrow areas further assessed the high quality of the lamellar matrix present inside the carbon cloth. By comparison, for the fibers located at a distance from the defect, the absence of mature bone formation in the interspace of the fibers strongly suggests that these carbon cloths have no intrinsic osteoinductive property and do not induce ectopic bone formation.

Finally, and importantly, no carbon fiber debris or carbon-bearing particles were found outside the implantation area as well as in the six major organs analyzed, namely the heart, liver, lungs, spleen, kidney and brain.

Altogether, these findings confirm the interest of using ACC scaffolds coated with CDA to accelerate bone reconstruction and demonstrate the beneficial effect of strontium CDA doping. This effect is correlated with an increase in the porous structure of the new bone providing a positive osteoconductive environment. CDA coating progressively dissolves to leave the fully osteointegrated carbon cloth without any detectable circulating carbon debris. In parallel, ACC scaffolds are progressively osteointegrated through the process of reconstruction.

## 5. Conclusions

In this study, the interest into CDA-coated ACC patches for bone reconstruction was highlighted and the beneficial effect of strontium doping of the CDA was demonstrated. Apposition of ACC/10Sr-CDA patches on a cortical bone defect allows for faster bone reconstruction and faster recovery of the initial shape of the femoral shaft. Moreover, Raman microspectroscopical data demonstrated the high quality of the newly formed bone with a chemical nature very close to the native bone. Follow-up at 6 months showed that these ACC cloths apposed on bone defects were fully osteointegrated in the reconstructed cortical bone, which demonstrates their high biocompatibility and osteoconductive properties. Finally, no traces of carbon-bearing debris were found in any of the six major organs examined. Owing to our previous results that described the adsorption/desorption properties of these carbon fibers and which demonstrated in vivo their ability to deliver therapeutic drugs in situ, we hypothesize that these new doped-CDA/ACC biomaterials might be useful as bone patches to accelerate bone repair in various clinical situations such as infection and cancer, which are important challenges in orthopedic and reconstructive surgery.

## Figures and Tables

**Figure 1 jfb-14-00246-f001:**
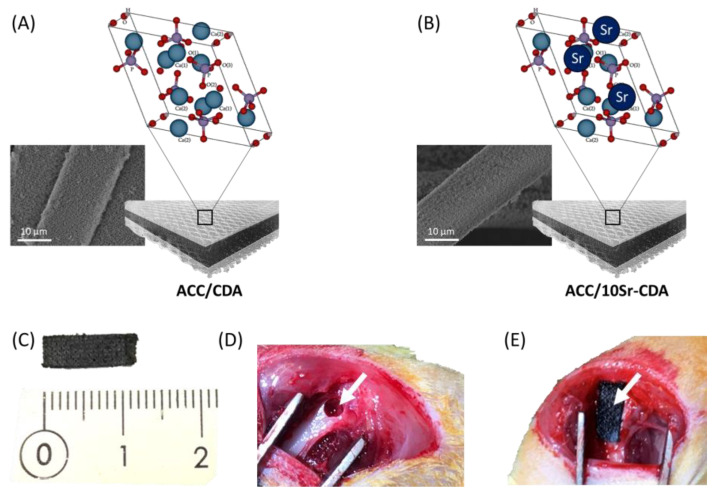
SEM images and schematic representation of the (**A**) ACC/CDA patch; and (**B**) ACC/10Sr-CDA patch. (**C**) Picture of a patch (10 × 4 mm); (**D**,**E**) pictures of a femoral defect (white arrow) before (**D**) and after (**E**) superficial apposition of a patch.

**Figure 2 jfb-14-00246-f002:**
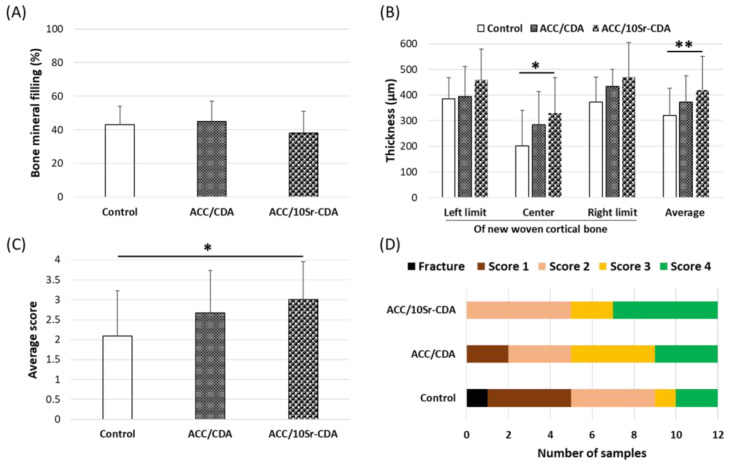
Analysis of bone defect healing at day 26 post-surgery. (**A**–**D**) µCT analysis; (**A**) quantification of bone mineral filling percentage; (**B**) quantification of new bone thickness; (**C**–**D**) bone shape scoring according to the scheme described in Table 2. Mean ± standard deviation, * *p* < 0.05 and ** *p* < 0.01.

**Figure 3 jfb-14-00246-f003:**
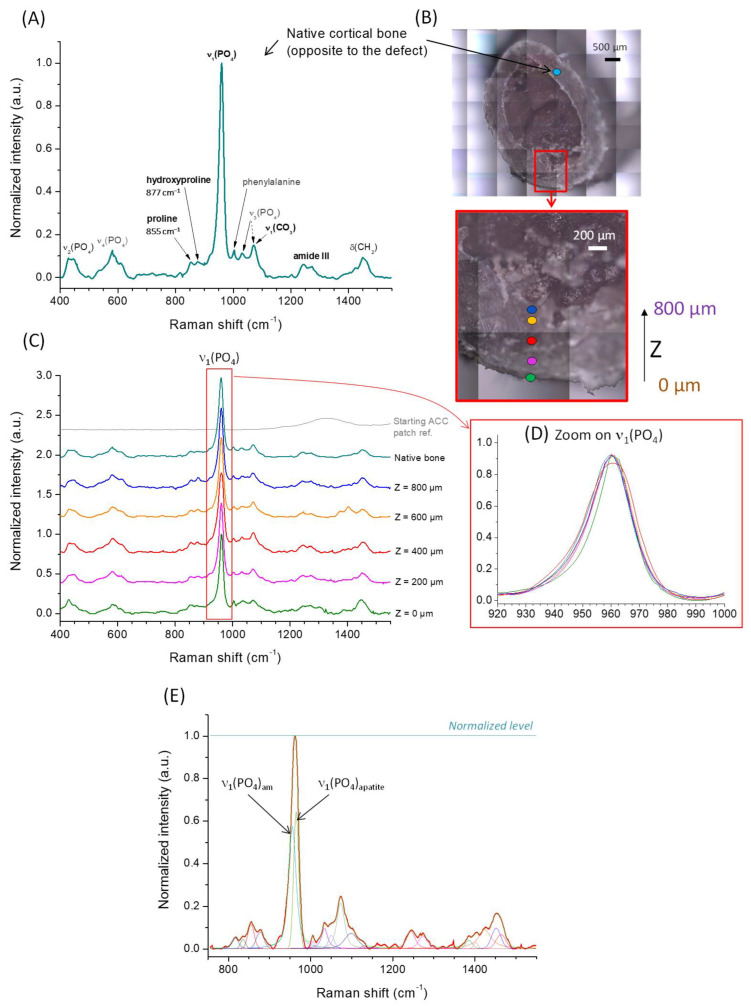
(**A**) Raman spectrum of native cortical bone (opposite to the defect) and main band assignments; (**B**) position of the analyzed domains along the depth (Z level from external to internal regions) of the newly formed bone after apposition of ACC/10Sr-CDA patch at day 26 post-surgery; (**C**) Raman spectra for different depths of analysis, for native bone and for the initial carbon cloth; (**D**) zoom on the ν_1_(PO_4_) spectral domain; (**E**) example of curve fitting for an analyzed depth of Z = 400 µm.

**Figure 4 jfb-14-00246-f004:**
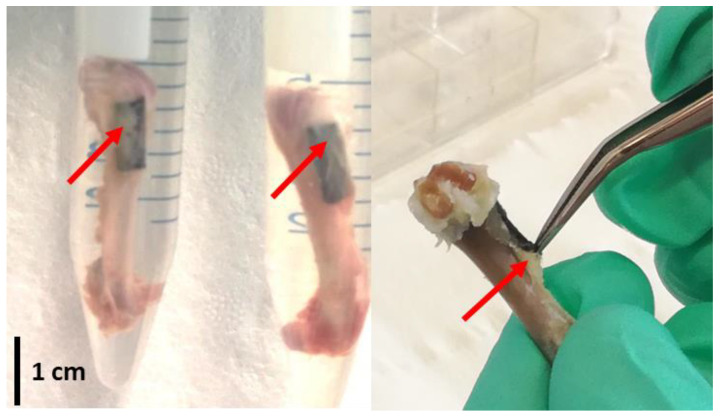
Pictures of patches attached to rat femur shafts, covered by a connective tissue (red arrow).

**Figure 5 jfb-14-00246-f005:**
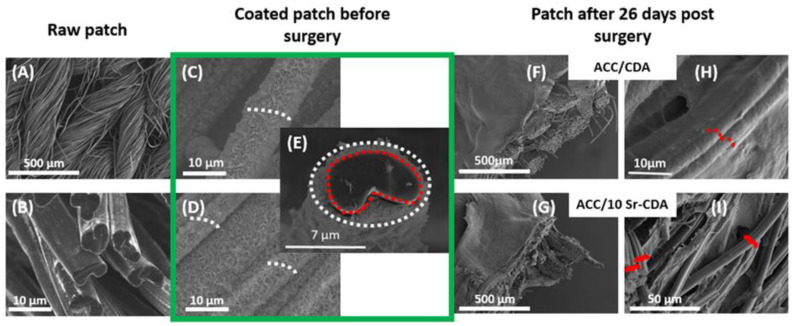
SEM images of: (**A**,**B**) raw ACC; (**C**–**E**) coated ACC before surgery, delineation of raw carbon (red dotted line) and CDA surface (white dotted line); (**F**–**I**) ACC/CDA patch (**F**,**H**) and ACC/10Sr-CDA patch (**G**,**I**) at day 26 post-surgery. In H and I a red dotted line delineates the raw carbon shape.

**Figure 6 jfb-14-00246-f006:**
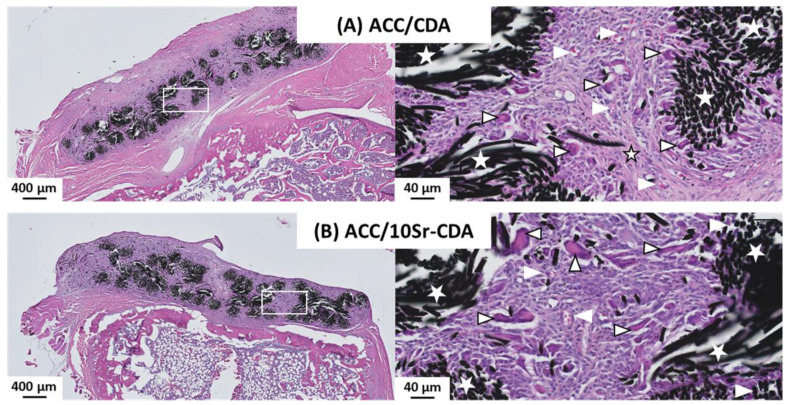
Fate of ACC/CDA and ACC/10Sr-CDA patches after 26 days of implantation. Optical microscopy examination of ACC/CDA (**A**) and ACC/10Sr-CDA (**B**) patches. Carbon fibers (white star), blood vessels (white arrowhead), giant cells (black arrowhead), and fibrillar collagenous tissue trabeculae (black star) are shown.

**Figure 7 jfb-14-00246-f007:**
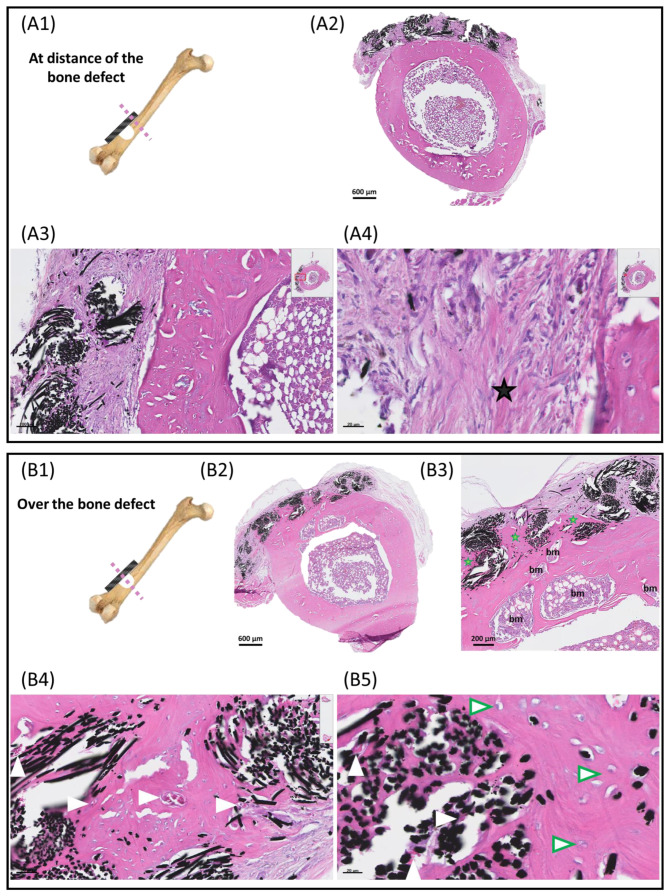
Fate of ACC patches after 6 months of implantation. Optical microscopy examination of a patch area at distance from the bone defect (**A**) and over the bone defect (**B**). Blood vessels (white arrowhead), collagenous trabeculae (black star), lamellar bone (green star), osteocytes (green arrowhead) and bone marrow areas (bm) are indicated.

**Figure 8 jfb-14-00246-f008:**
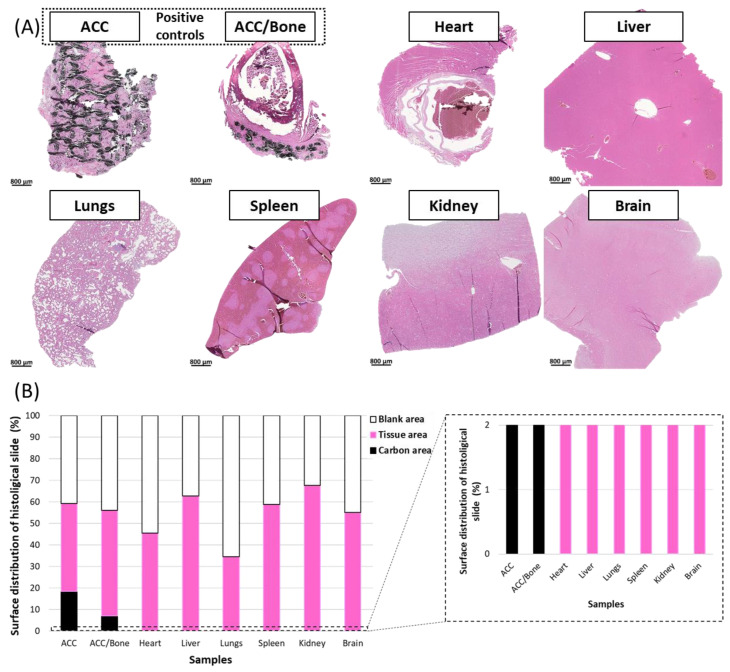
High resolution detection of carbon using the HALO Image Analysis Platform. (**A**) Pictures of optical microscopy examination of ACC after implantation and ACC/Bone slides used as positive controls and of major peripheral organs (heart, liver, lungs, spleen, kidney and brain) taken from animals treated with ACC patches at 6 months post-surgery. (**B**) Quantification of carbon (black), tissue (pink) and blank (white) in the analyzed areas. Carbon was never detected in the peripheral organs.

**Table 1 jfb-14-00246-t001:** First experiment: Repartition of the 36 femurs in the 3 groups, namely control, ACC/CDA, ACC/10Sr-CDA. Bone reconstruction was analyzed at day 26 post-surgery.

Groups	Number of Samples
Control	12
ACC/CDA	12
ACC/10Sr-CDA	12

**Table 2 jfb-14-00246-t002:** Scoring scheme for µCT evaluation of bone reconstruction shape. The green arrow indicates the initial (transpiercing) bone defect made.

Score	Associated Finding at Defect Site	µCT Image
1	Incomplete closure with persistent hole	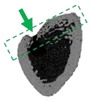
2	Complete closure with persistent concavity	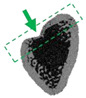
3	Complete closure without concavity	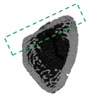
4	Native bone shape	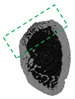

## Data Availability

All the data used and/or analyzed for the current study are contained in the article. All other datasets are available from the corresponding author upon reasonable request.

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
