# Peer review of "Long-Term Fate and Efficacy of a Biomimetic (Sr)-Apatite-Coated Carbon Patch Used for Bone Reconstruction"

_jfb, 2023, doi:10.3390/jfb14050246_

Round 1

Reviewer 1 Report

-Improve the introduction with data about other ways for improvement of implants. This references are suggested: https://doi.org/10.3390/mi12121447

-  Add more new references (2022).   

-Why you choose the coating? you can add a comparison with advantages and disadvantages compared to other used covers.

-The authors do not make at all clear what the novel aspect of their work is. Improve this. You can make a graph in the introduction with an overview to better highlight the structure of the work.

- Also please indicate in Introduction other studies already been done.

- Please discuss in Introduction about the applications.

- Generally the quality of the writing could be improved.

Author Response

Dear Reviewer,

We thank you very much for reviewing our manuscript. We have addressed each of your comments and you will find our answers below:

Comments and Suggestions for Authors :

-Improve the introduction with data about other ways for improvement of implants. This references are suggested: https://doi.org/10.3390/mi12121447

- Add more new references (2022).  

- Why you choose the coating? you can add a comparison with advantages and disadvantages compared to other used covers.

- Also please indicate in Introduction other studies already been done.

Answer:

We thank you for these comments. As asked, the following modifications have been made:

. The reference Baltatu et al., is now added in the introduction as well another article of the same group. Please see ref 27 and ref 30.

. New recent references have also been added in the manuscript [please see ref 12, 26-31 and 43-45].

. Other used covers and studies already done have been cited in the introduction as follows (see p2, line 53):

“In addition to their use as bone fillers, biomimetic CaP compounds can be used as coatings to improve the osteointegration of various implant types [26]. Several methods have been proposed to deposit such bioactive coatings such as atmospheric plasma spraying (APS), physical and chemical vapor deposition (PVD and CVD) [27], sol-gel [28], electrodeposition [29] or by immersion in a biomimetic solution [30]. For porous materials (3D), electrodeposition has been described among the most promising methods since it can deposit homogeneous bioactive coatings such as CaP both on the interior and exterior surfaces of conductive scaffolds, regardless of sample size and pore geometry [31].

 In this line we have previously demonstrated that the sono-electrodeposition process allowed obtaining a biomimetic calcium-deficient apatite (CDA) coating on activated carbon fiber cloth (ACC) to be used as bone patches [32]. […].”

-The authors do not make at all clear what the novel aspect of their work is. Improve this. You can make a graph in the introduction with an overview to better highlight the structure of the work.

- Please discuss in Introduction about the applications.

Answer:

To answer to these two comments, we have clarified the introduction section as follows (see p2, line 75):

“The goal of the present study was threefold. Our first objective was to investigate the efficacy of such composite carbon fiber patches in vivo at medium-term (26 days post-surgery). Using a model of critical bone defect in rat femur, we analyzed bone reconstruction in the presence or in the absence of patches of ACC/CDA or the strontium-doped equivalent ACC/10Sr-CDA. This allowed to test the interest of strontium doping. The second objective of this work was to analyze the medium and long term fate of these patches (26 days and 6 months after surgery), i.e. their biocompatibility and osteointegration. The third objective was to answer the question whether carbon debris could be found out of the implantation site and within peripheral organs. Analyses of ACC and CDA coatings (undoped or Sr2+-doped) were carried out by microscopy techniques (optical and scanning electron microscopy, SEM). The quality of the newly-formed bone was checked using Raman microspectroscopy through the analysis of different quantitative parameters (mineral-to-organic ratio, apatite maturity state, carbonation level, hydroxyproline-to-proline ratio) and compared to those obtained with native cortical bone.”  

- Generally the quality of the writing could be improved.

Answer:

We have checked the quality of the writing.

Reviewer 2 Report

In their 19-page manuscript entitled: "Long term fate and efficacy of biomimetic apatite-coated car-bone patch used for bone reconstruction", the authors present an interesting approach to the treatment of bone defects.

Unfortunately, there are some major errors in the manuscript, so I recommend a major revision before publication.

In the following, my comments are listed chronologically, as there are no line numbers in the manuscript:

2.1 Materials

·         All materials, as well as their origin, which were used in the later staining are missing.

·         The headquarters and country of the Zorflex Company are missing.

·         What does “thorough” mean in time specifications? With regard to reproducibility Seconds? Minutes?

2.2 Study Design

·         why were only male animals used?

·         figure 1C is not really meaningful, I would enlarge it

2.3 Surgical procedure

·         Which x-ray device was used? Which settings?

2.5 µCT

·         CT software (Bruker) … Headquarter, country, version used are missing

·         Matlab software … Manufacturer, headquarter, country and version used are missing

·         was the "bone mineral filling %" calculated in 2D using individual sectional images from the CT or the rendered volume body from the CT data in 3D?

2.6 Statistical analysis

·         more significant than significant does not exist mathematically ;)

2.7 SEM

·         how thick was the sputter-coated gold layer?

·         (SEM-Hitachi S4500)… Headquarter, country missing

2.8 Raman

·         Raman Labram HR 800 Horiba Jobin Yvon microscope … Manufacturer, headquarter, country missing

·         Olympus BX 41 … Manufacturer, headquarter, country missing … it should be like this: “…Olympus BX 41 (Olympus Inc., Tokyo, Japan) …”

·         XYZ motorized table … Manufacturer, headquarter, country missing

2.9

·         EDTA … missing under materials, also the indication where it was obtained from

·         (HE) … missing under materials, also the indication where it was obtained from

Results

3.1

·         “…ACC/CDA, ACC/10Sr-CDA and control, Figure 2A …” - Here I would also like to see a µCT image in which the new mineralization is marked. If the authors already make µct recordings why don't they present them in the manuscript? similar to the schematic representations of the classification of the defect areas in the µCT into a score in tab.2

·         “…This appears to be linked to a higher porosity of the newly-formed bone…” - how high was the porosity, informations missing

·         “…normalized with reference to the apatite peak denoted “v1(PO4)” at ca. 960 cm-1 and reported on Figure 3C….” (and following paragraph with several ca.; ~; about) please do not give approximate but exact values - this is a scientific paper not a newspaper article. Please measure the exact value (as mean ± SD like 960 ± 7 cm-1)

·         Figure 3 … Please adjust the colors and avoid red/green (for readers with red-green visual impairment there is no difference)

·         “…Beside changes observed in the mineral phase, the organic component of the newly- formed bone tissue may also be examined as collagen can also undergo modifications upon aging/neoformation, which may lead to relevant changes in Raman spectra …” - are there altered mechanical parameters of the bone due to these changes in the bone? Compressive strength/tensile strength, if applicable, changed Young´s modulus?

3.2.2

·         “…were fully colonized by an active fibroblastic tissue, including fibrillar col- lagen fibers, many fibroblastic cells, blood vessels and giant multinucleated cells…” - could these not be foreign body giant cells, indicating a foreign body reaction of the immune system

3.2.4

·         and the carbon fibers could not have fallen out due to a systematic error when cutting on the microtome (as they are harder than the surrounding tissue)?

5. Conclusions

The authors conclude that the CDA-coated ACC patches induce the formation of new bone with highly porous architecture, but do not present data on the measured porosity. Please add a paragraph about porosity measurement and a few more images SEM / µCT / Origin graphs about porosity.

Author Response

Dear Reviewer,

We thank you very much for reviewing our manuscript. We have addressed each of your comments and you will find our answers below:

Comments and Suggestions for Authors

In their 19-page manuscript entitled: "Long term fate and efficacy of biomimetic apatite-coated car-bone patch used for bone reconstruction", the authors present an interesting approach to the treatment of bone defects.

Unfortunately, there are some major errors in the manuscript, so I recommend a major revision before publication.

In the following, my comments are listed chronologically, as there are no line numbers in the manuscript:

Answer:

We apologize but we have loaded the article including line numbers, we guess this is a technical problem.

2.1 Materials

  • All materials, as well as their origin, which were used in the later staining are missing.

Answer:

We apologize for these omissions, please see the modifications made at point 2.9.

  • The headquarters and country of the Zorflex Company are missing.

Answer:

The headquarters and country have been added as follows (p2, line 93-94): “(Zorflex® ACC, Calgon Carbon Corporation, Pittsburgh, Pennsylvania, US).”

  • What does “thorough” mean in time specifications? With regard to reproducibility Seconds? Minutes?

 Answer:

The sentence has been modified as follows (p2, line 94-95):  “Patches of 10 x 4 mm were cut and rinsed cautiously in deionized water for 15 seconds to remove any undesirable carbon residue”.

2.2 Study Design

  • why were only male animals used?

Answer:

We have made these experiments on male only first because all our experiments for many years have been done in males, which allows to compare our results from one experiment to another; second in order to reduce the number of used animals. Indeed using males and females would require to duplicate the number of experiments; third since at the same age male have larger bones than females, which facilitates surgery.

  • figure 1C is not really meaningful, I would enlarge it

Answer:

As asked, Figure 1C has been enlarged.

2.3 Surgical procedure

  • Which x-ray device was used? Which settings?

Answer:

We have used a Faxitron x-ray machine (Edimex, Le Plessis Grammoire, France) at a voltage of 35 kV and exposure time of 9.5 s.

This has been added in the text at 2.3 Surgical procedure as follows (p4, line 136 to 138): “A postoperative X-Ray (Faxitron, Edimex, Le Plessis Grammoire, France) check was performed - at voltage of 35 kV and exposure time of 9.5 s - to assess the absence of fracture”.

2.5 µCT

  • CT software (Bruker) … Headquarter, country, version used are missing

Answer:

Headquarter, country, version used have been added (p4 line 154-155)

“The CTAn software (1.17.7.2 + version, Bruker microCT, Billerica, Massachusetts, US) was used…”

  • Matlab software … Manufacturer, headquarter, country and version used are missing

Answer:

The company, headquarters and country have been added. (p4 line 157)

“The Matlab software (R2017b version, The MathWorks Inc., Natick, Massachusetts, USA)…”

  • was the "bone mineral filling %" calculated in 2D using individual sectional images from the CT or the rendered volume body from the CT data in 3D?

Answer:

We have added this in material and methods 2.5: (p4 line 158 to 162).

“The region containing the bone defect was defined inside a constant volume cylinder (3D). Each 2D slice of this volume was then binarized individually to select the newly formed mineral. The resulting binary images, acting as masks were multiplied by their own slices to finally calculate the volume of the newly formed mineral.”

2.6 Statistical analysis

  • more significant than significant does not exist mathematically ;)

Answer:

The paragraph dealing with statistical evaluation has been modified as follows: “Statistical evaluation of data was performed using Student test. The experiment was repeated twelve times and the results presented are the mean of the values. Data are reported as ± standard deviation at a significance level of at *p < 0.05 and **p < 0.01.”  (p5, line 178 to 180)

2.7 SEM

  • how thick was the sputter-coated gold layer?

  • (SEM-Hitachi S4500)… Headquarter, country missing

Answer:

We have now added the missing information (p5 line 184 to 186).

“They were then mounted on aluminum stubs and sputter coated with gold 2 to 3 nm gold layer. Examination was performed using a field emission scanning electron microscope (SEM-Hitachi S4500, Hitachi High-Tech Corporation, Tokyo, Japan) operating at 5 kV.”

2.8 Raman

  • Raman Labram HR 800 Horiba Jobin Yvon microscope … Manufacturer, headquarter, country missing

  • Olympus BX 41 … Manufacturer, headquarter, country missing … it should be like this: “…Olympus BX 41 (Olympus Inc., Tokyo, Japan) …”

  • XYZ motorized table … Manufacturer, headquarter, country missing

Answer:

We have now added the missing information. (see p5-6 line 194, 198, 206)

  • a confocal Raman Labram HR 800 Horiba Jobin Yvon microscope (Horiba FRANCE SAS, Palaiseau, France)
  • an Olympus BX 41 microscope (Olympus Europa SE & CO, Hamburg, Germany)
  • a XYZ motorized table (Märzhäuser GMBH & Co., Wetzlar, Germany)

2.9

  • EDTA … missing under materials, also the indication where it was obtained from

Answer:

This information has been added: ethylenediaminetetracetic acid (EDTA, Alfa Aesar, Thermo Fisher Scientific, Illkirch, France). (p6 line 218)

  • (HE, ) … missing under materials, also the indication where it was obtained from

Answer:

Hematoxylin (Harris Hematoxylin solution, Sigma-Aldrich, Saint Quentin Fallavier, France) Eosin (Eosin 0,5% alcoholic, Diapath, Voisins-le-Bretonneux, France). (p6 line 222)

Results

3.1

  • “…ACC/CDA, ACC/10Sr-CDA and control, Figure 2A …” - Here I would also like to see a µCT image in which the new mineralization is marked. If the authors already make µct recordings why don't they present them in the manuscript? similar to the schematic representations of the classification of the defect areas in the µCT into a score in tab.2

Answer:

Figure 2A shows a quantitative volume analysis (3D) performed using Matlab computational software (see answer 2.5 µCT). This cannot be represented by a 2D image like in Table 2. In Table 2, these are 2D images that score the surface morphology of the bone.

  • “…This appears to be linked to a higher porosity of the newly-formed bone…” - how high was the porosity, informations missing

Answer:

We were not able to make an accurate measurement of the porosity of the newly formed bone. The "porosity" referred to here is deduced from the interpretation of the data of mCT analysis shown in the figure 2. As indicated in the results 3.1, from these data it can be deduced that, in the presence of ACC/CDA and more particularly of ACC/10Sr-CDA patches, the ratio between the total volume of new bone (bone thickness shown in fig 2B) and total amount bone mineral (shown in fig 2A) is increased. This strongly suggests that new bone formed in the presence of these patches has a more porous structure compared to new bone formed in the absence of patch.

To take into account your comment, we have modified the text as follows (p6, end of the first paragraph of the section 3.1, line 246):

This appears to be linked to a higher porosity of the newly-formed bone. “This strongly suggests that the bone formed in contact with the strontium-doped patches has a larger porous structure than that of the bone formed in the other two groups.”

We have also modified the discussion section as follows (p15, second paragraph, line 482 to 487)

“Despite the fact that we could not make an accurate measurement of porosity, we can make the hypothesis that new bone formed in the presence of a Sr2+-doped patches has a higher porous structure than new bone formed in the presence of undoped patches or in the absence of patch. This higher porous structure should likely provide a better inter-connection supporting cell migration, neo-vascularization, new tissue ingrowth and bone healing.”

  • “…normalized with reference to the apatite peak denoted “v1(PO4)” at ca. 960 cm-1 and reported on Figure 3C….” (and following paragraph with several ca.; ~; about) please do not give approximate but exact values - this is a scientific paper not a newspaper article. Please measure the exact value (as mean ± SD like 960 ± 7 cm-1)

Answer:

In fact, we deliberately mentioned the “approximate” position of the normalizing v1(PO4) band because its position differs from one sample to the other, due to several reasons including the differing degree of crystallinity and the presence of two underlying contributions, as we mention in the text. This is why we gave this “ca. 960 cm-1” position. However, please note that in each case, the exact value (and not the approximation) of the peak maximum was taken into account for the plotting. This was the same for the other bands. We also remind that we gave all the bands exact positions in the complete table found in Supplementary Information… But in order to take into account the Reviewer’s preference, we have now mentioned in the form “mean ± SD” for all Raman band positions.

  • Figure 3 … Please adjust the colors and avoid red/green (for readers with red-green visual impairment there is no difference)

Answer:

We were unfortunately limited in the number of available colors, but we have indicated clearly the value of Z in the figure so that there is no need for distinguishing colors, the figures are understandable even in black and white.

  • “…Beside changes observed in the mineral phase, the organic component of the newly- formed bone tissue may also be examined as collagen can also undergo modifications upon aging/neoformation, which may lead to relevant changes in Raman spectra …” - are there altered mechanical parameters of the bone due to these changes in the bone? Compressive strength/tensile strength, if applicable, changed Young´s modulus?

Answer:

We may indeed expect some variability in local mechanical properties, but we did not have any way to measure it at this scale. We already stated in the text on page 8 that “the mechanical properties of the bony matrix were found to be directly linked to this mineral-to-organic ratio.” Also, since the porosity also differs during this stage of new-bone formation, the mechanical properties of the tissue likely has local variability. But our results point out a noticeable level of new bone formation, and the cells thus seem to not be negatively affected by the local modifications of mechanical stiffness. To take into account nonetheless this comment, we have now added a 5-line paragraph on page 10 (line 374) as follows:

“We expect local variation in the mechanical properties of the newly formed bone tissue, due in part to the porosity of the new bony matrix in formation, and also to local variations in mineral-to-organic ratio among other parameters. Evaluation at the micro-scale of mechanical properties by approaches like Brillouin-Raman microspectroscopy could prove helpful to explore further, in the future, this type of new bone tissue [43-45].”

3.2.2

  • “…were fully colonized by an active fibroblastic tissue, including fibrillar col- lagen fibers, many fibroblastic cells, blood vessels and giant multinucleated cells…” - could these not be foreign body giant cells, indicating a foreign body reaction of the immune system.

Answer:

We agree with the reviewer but in this context of bone formation, these giant multinucleated cells are more likely osteoclast-like cells. The presence of these cells is consistent with the first phase of bone formation, which is observed in the presence of all biomaterials implanted in vivo, and is the preliminary stage to bone formation by osteoblasts. This is consistent with the evolution that we observed after 6 months, i.e. the formation of mature bone within the carbon fibers.

We have added a sentence in the discussion section as follows: “At both times, neo-vascularization was a major indicator of their good integration as well as the presence of giant multinucleated cells. These cells are more likely osteoclast-like cells whose presence is consistent with the first phase of bone formation by osteoblasts, as observed in the presence of all biomaterials implanted in vivo.”  (p16, line 526)

3.2.4

  • and the carbon fibers could not have fallen out due to a systematic error when cutting on the microtome (as they are harder than the surrounding tissue)?

Answer:

We are not sure to understand this comment of the reviewer. Carbon fibers are slightly harder than the surrounding tissue but this did not induce any artefact during cutting. Here, the presence of carbon was observed only in the positive controls. 

  1. Conclusions

The authors conclude that the CDA-coated ACC patches induce the formation of new bone with highly porous architecture, but do not present data on the measured porosity. Please add a paragraph about porosity measurement and a few more images SEM / µCT / Origin graphs about porosity.

Answer:

We hope that our previous responses can answer to this last comment. For the reasons explained above (please see our answers to results 3.1) we cannot provide neither with porosity measurement nor with SEM or µCT images.

Nevertheless we totally agree that the sentence “formation of new bone with a highly favorable porous architecture”  is confusing since the term “highly” was supposed to concern the term “favorable” and not to the term “porous”.

As indicated above we have modified the discussion section as follows (p15, second paragraph, line 482)

“Despite the fact that we could not make an accurate measurement of porosity, we can make the hypothesis that new bone formed in the presence of a Sr2+-doped patches has a higher porous structure than new bone formed in the presence of undoped patches or in the absence of patch. This higher porous structure should likely provide a better inter-connection supporting cell migration, neo-vascularization, new tissue ingrowth and bone healing.”

Finally we have also modified our conclusion as follows: (p16, line 549)

” Apposition of ACC/10Sr-CDA patches on a cortical bone defect allows for faster bone reconstruction and faster recovery of the initial shape of the femoral shaft. Interestingly, the effect of these patches is likely related to an increase of the porous structure of the new bone. Moreover, raman microspectroscopical data demonstrated the high quality of the newly formed bone with a chemical nature very close to the native bone.”.

Reviewer 3 Report

The study entitled “Long term fate and efficacy of biomimetic apatite-coated carbon patch used for bone reconstruction” aims at investigating over longer time periods the reconstruction of cortical bone in the presence of activated carbon cloths with calcium-deficient apatites or strontium-doped calcium-deficient apatites patches, as well as the behavior of the carbon fibers.

The paper is well-structured, with clarity in ideas and demonstrations, and the methodology part coherently expressed. Nevertheless, throughout the manuscript there are some points that need a slight improvement before its acceptance.

Therefore, some comments and suggestions are listed below.

1. In this paper, as reported in the Conclusions section, the interest of CDA-coated ACC patches for bone reconstruction was highlighted and the fate of these biomaterials after mid- and long-term implantation was studied. However, authors should mainly highlight the role of strontium doping both in the title and throughout the manuscript since this coating was found to be the best result in this study, thus improving quantitatively the new bone formation without altering its chemical composition.

2. At the end of the introduction section, authors assert that “the quality of the newly formed bone was checked using Raman micro-spectroscopy through the analysis of different quantitative parameters and compared to those obtained with native cortical bone.“ Since Raman spectroscopy is a powerful molecular tool which can track molecular changes at high spatial resolution giving information on mineral-to-organic ratio, apatite maturity state, carbonation level, hydroxyproline-to-proline ratio, this vibrational technique should be better described. Although in the Discussion section authors provide a more detailed description, they only cite the work by Paschalis et al. (#Ref. 33) that is a general overview of vibrational spectroscopic techniques to assess bone quality. On the other hand,   recent works by Alunni Cardinali et al. [(https://doi.org/10.1038/s41598-020-74330-3), (https://doi.org/10.1098/rsif.2021.0642), (https://doi.org/10.3390/ma14226869)] better describe the detection of molecular change in specific part of femoral bone-i.e., the same anatomical region analysed by the authors. Therefore, these references may help authors to better support their sentences.

3. In the last paragraph of Conclusion section, authors assert that “this study allowed us to shed light on longer-term in vivo implantation of such CDA-coated carbon cloths, and is expected to help develop in the future some translational applications to the clinic.” This part should be slight improved also adding appropriate references which describe future potential applications of the mentioned biomaterials.

4. Please note that Supplementary material (i.e., Figure S1 and Table S1 mentioned in the text) was not uploaded during the submission process.

Author Response

Dear Reviewer,

We thank you very much for reviewing our manuscript. We have addressed each of your comments and you will find our answers below:

Comments and Suggestions for Authors

The study entitled “Long term fate and efficacy of biomimetic apatite-coated carbon patch used for bone reconstruction” aims at investigating over longer time periods the reconstruction of cortical bone in the presence of activated carbon cloths with calcium-deficient apatites or strontium-doped calcium-deficient apatites patches, as well as the behavior of the carbon fibers.

The paper is well-structured, with clarity in ideas and demonstrations, and the methodology part coherently expressed. Nevertheless, throughout the manuscript there are some points that need a slight improvement before its acceptance.

Therefore, some comments and suggestions are listed below.

  1. In this paper, as reported in the Conclusions section, the interest of CDA-coated ACC patches for bone reconstruction was highlighted and the fate of these biomaterials after mid- and long-term implantation was studied. However, authors should mainly highlight the role of strontium doping both in the title and throughout the manuscript since this coating was found to be the best result in this study, thus improving quantitatively the new bone formation without altering its chemical composition.

Answer:

We agree with your comments and we have made the following changes:

1) the title has been modified as follows : “Long term fate and efficacy of biomimetic (Sr)-apatite-coated carbon patch used for bone reconstruction”

2) The role of strontium doping has been highlighted throughout the manuscript, abstract, discussion and conclusion.

Abstract: (p1, lines 20-25)

This study aimed to analyze at medium term the reconstruction of cortical bone in the presence of ACC/CDA or ACC/10Sr-CDA patches corresponding to 6 at.% of strontium substitution. It also aimed to examine the behavior of these cloths at medium and long term, in situ and at distance. Our results at day 26 confirm the particular efficacy of strontium-doped patches on bone reconstruction, leading to new thick bone with high bone quality as quantified by Raman microspectroscopy.

Discussion (p15, line 475 to 495):

 “ Our in vivo data in rat at day 26 post-surgery clearly indicate that the apposition of these composite patches improves the bone healing process more importantly in the presence of a strontium-doped CDA coating. These data confirm the guiding effect of the ACC fiber patches that we described previously [34] as well as the osteogenic effect of strontium cations described in many studies [18-20]. Moreover we also showed that this increase of new bone volume in the presence of strontium occurred although the amount of new formed mineral matrix was similar in the three groups namely control, ACC/CDA and ACC/10Sr-CDA. Despite the fact that we could not make an accurate measurement of porosity, we can make the hypothesis that new bone formed in the presence of a Sr2+-doped patches has a higher porous structure than new bone formed in the presence of undoped patches or in the absence of patch. This higher porous structure should likely provide a better interconnection supporting cell migration, neo-vascularization, new tissue ingrowth and bone healing. Our results further showed that this significant augmentation of bone volume in the presence of Sr2+ was related to an increase of thickness at the center of the reconstructed bone. This result is consistent with our data of bone shape scoring showing that ACC/10Sr-CDA patches apposition led to the highest number of femur recovering their native convex shape. This is also consistent with a recent work of Quade et al. relative to the boosting role of Sr2+ ions on BMP-2 mediated bone regeneration of mice femurs [25]. In this study, the authors concluded, by means of morphological scores, that the quality of the newly-formed bone was significantly en-hanced in Sr-modified scaffolds in a murine segmental bone defect model.”

Modifications has been made also at the end of the discussion (line 539), as follows :

“Altogether these findings confirm the interest of using ACC scaffolds coated with CDA to accelerate bone reconstruction and demonstrate the beneficial effect of strontium CDA doping.”

  1. At the end of the introduction section, authors assert that “the quality of the newly formed bone was checked using Raman micro-spectroscopy through the analysis of different quantitative parameters and compared to those obtained with native cortical bone.“ Since Raman spectroscopy is a powerful molecular tool which can track molecular changes at high spatial resolution giving information on mineral-to-organic ratio, apatite maturity state, carbonation level, hydroxyproline-to-proline ratio, this vibrational technique should be better described. Although in the Discussion section authors provide a more detailed description, they only cite the work by Paschalis et al. (#Ref. 33) that is a general overview of vibrational spectroscopic techniques to assess bone quality. On the other hand, recent works by Alunni Cardinali et al. [(https://doi.org/10.1038/s41598-020-74330-3), (https://doi.org/10.1098/rsif.2021.0642), (https://doi.org/10.3390/ma14226869)] better describe the detection of molecular change in specific part of femoral bone-i.e., the same anatomical region analysed by the authors. Therefore, these references may help authors to better support their sentences.

Answer:

We thank the Reviewer for this comment. We have now cited these three other references on page 7 and page 10, and added a 5-line paragraph on page 10 as follows:

“We expect local variation in the mechanical properties of the newly formed bone tissue, due in part to the porosity of the new bony matrix in formation, and also to local variations in mineral-to-organic ratio among other parameters. Evaluation at the micro-scale of mechanical properties by approaches like Brillouin-Raman microspectroscopy could prove helpful to explore further, in the future, this type of new bone tissue [42-44].” (p10 line 374 to 379).

  1. In the last paragraph of Conclusion section, authors assert that “this study allowed us to shed light on longer-term in vivo implantation of such CDA-coated carbon cloths, and is expected to help develop in the future some translational applications to the clinic.” This part should be slight improved also adding appropriate references which describe future potential applications of the mentioned biomaterials.

Answer:

The conclusion has been improved to answer to this comment, as follows (p 16, line 547):

“In this study, the interest of CDA-coated ACC patches for bone reconstruction was highlighted and the beneficial effect of strontium doping of the CDA was demonstrated. Apposition of ACC/10Sr-CDA patches on a cortical bone defect allows for faster bone reconstruction and faster recovery of the initial shape of the femoral shaft. Moreover, raman microspectroscopical data demonstrated the high quality of the newly formed bone with a chemical nature very close to the native bone. Follow-up at 6 months showed that these ACC cloths apposed on bone defects were fully osteointegrated in the reconstructed cortical, which demonstrates their high biocompatibility and osteoconductive properties. Finally no traces of carbon-bearing debris were found in any of the 6 major organs examined. Owing to our previous results that described the adsorption/desorption properties of these carbon fibers and which demonstrated in vivo their ability  to deliver therapeutic drugs in situ, we guess that these new doped-CDA/ACC biomaterials might be useful as bone patches to accelerate bone repair in various clinical situations such as infection and cancer, which are important challenges in orthopedic and reconstructive surgery.”

  1. Please note that Supplementary material (i.e., Figure S1 and Table S1 mentioned in the text) was not uploaded during the submission process.

Answer:

We apologize for this problem that we don’t understand since the Supplementary material file was uploaded during the submission process.

Round 2

Reviewer 1 Report

Paper was significant improved. 

Reviewer 2 Report

In my opinion, the revised version of the manuscript can be accepted for publication in its present form .

Reviewer 3 Report

The authors have addressed the issues raised previously, and the manuscript is suitable for publication in its current form.